# Integrated multifactor assessment of road geometry, vehicle types and weather diversity on bilateral transverse slopes: Bridging gaps in dynamic modeling

Mahdi Moharami[1], Ali Abdi Kordani[1]*, Akram Kohansal[2], Farzad Moradi[1]

1 Department of Civil–Transportation Planning, Imam Khomeini International University, Qazvin, Iran,
2 Department of Statistics, Imam Khomeini International University, Qazvin, Iran

* aliabdi@eng.ikiu.ac.ir

## Abstract

Lane-changing maneuvers are critical to road safety and are significantly influenced by road geometry, particularly transverse slopes. This study investigates how bilateral transverse slopes affect vehicle dynamics during lane changes and their implications for safety. Using advanced simulation tools, CarSim and TruckSim, a comprehensive model was developed incorporating vehicle types, speeds, weather conditions, and driver behaviors. The analysis covered three vehicle categories—sedans, SUVs, and trucks—across four transverse slope variations (0%, 1.5%, 2%, and 2.5%) under three weather conditions: dry, rainy, and snowy. Key dynamic parameters such as steering angle, slip angle, rollover angle, and torsional moment angle were examined to evaluate each vehicle's stability and behavior throughout lane-changing maneuvers. The results show that increasing transverse slopes moderately affect steering and rollover angles, while slip angles and torsional moments remain largely stable for all vehicle types. Trucks demonstrated the highest sensitivity to speed and slope changes, with control challenges arising at speeds above 90 km/h on higher slopes. In contrast, sedans and SUVs exhibited stable behavior with minimal variations in the safety parameters measured. Additionally, adverse weather conditions—especially snowy roads characterized by low friction coefficients (0.28)—amplify instability risks across all vehicles. In conclusion, bilateral transverse slopes, primarily implemented for drainage purposes, can be safely incorporated into suburban road designs with slopes up to 2.5% without significantly compromising vehicle stability. These findings offer important insights for optimizing road geometry, enhancing vehicle safety systems like Electronic Stability Control (ESC), and reducing rollover risks under a variety of driving conditions.

**Data availability statement:** All relevant data are within the paper and its Supporting information files.

**Funding:** The author(s) received no specific funding for this work.

**Competing interests:** The authors have declared that no competing interests exist.

## Introduction

Lane-changing maneuvers are pivotal moments in traffic dynamics that critically influence road safety by interacting with vehicle behavior and road infrastructure features. Among these features, road geometry—particularly bilateral transverse slopes—plays a complex role in shaping vehicle stability during such maneuvers. Though primarily designed for effective drainage, bilateral transverse slopes introduce asymmetrical forces that affect lateral and vertical vehicle accelerations, as well as rollover and yaw tendencies. These effects become pronounced under varying weather conditions, especially adverse environments like rain and snow, where reduced tire-road friction further complicates vehicle control. This multifactor interplay remains understudied despite its significance, particularly regarding a comprehensive evaluation of vehicle types, speed ranges, slope severities, and weather influences simultaneously [1,2].

Although numerous studies have examined the influence of curve radius, superelevation, and surface friction on vehicle stability, most have focused on unidirectional gradient or standard superelevated curves, rather than on bilateral transverse slopes used on straight or quasi-straight suburban segments. In practice, many suburban roads employ bilateral transverse slopes primarily for drainage, creating opposite slopes between lanes that can alter lateral load transfer during lane changes, especially for heavy vehicles and under rain or snow. Yet current design guidelines largely extrapolate from curve-based superelevation research, providing limited quantitative evidence on how specific bilateral slope levels (for example 1.5–2.5 percent) interact with speed, vehicle type, and weather in dynamic maneuvers such as lane changes.

Recent studies have highlighted the increased risks posed by transverse slope gradients on heavy vehicles, such as trucks and SUVs, with rollover likelihoods markedly rising on icy or highly sloped roads exceeding 2% transverse inclination [3–6]. While research has elucidated isolated impacts of road texture improvements and slope gradients on traction [7], there remains a paucity of studies that integrate multiple vehicle categories and dynamic environmental scenarios within a unified simulation framework. Compounding factors, including interaction effects of speed, slope degrees, vehicle mass center, and weather-related road friction, demand high-fidelity modeling to capture real-world distribution of safety hazards. The present study confronts these gaps by leveraging advanced simulation platforms, CarSim and TruckSim, to analyze lane-changing maneuvers across sedans, SUVs, and trucks under slopes varying from 0% to 2.5% and speeds between 80 and 120 km/h in dry, rainy, and snowy conditions [1,8–11].

Prior literature broadly acknowledges the influence of road geometry on vehicle lateral stability. Cross-section characteristics such as curvature, camber, and especially transverse slopes modify key vehicle dynamic indicators including lateral acceleration, slip angle, and roll angle, which are decisive in accident causation and severity [12–15]. The presence of adverse weather exacerbates instability risks by decreasing tire-road friction coefficients from approximately 0.9 in dry conditions to as low as 0.28 on snowy surfaces, intensifying rollover and sideslip propensities [16,17]. Speeds above 90 km/h have been consistently linked to elevated rollover risks,

particularly among heavy vehicles with high centers of gravity [18,19]. Despite these insights, existing studies often isolate variables or focus on urban slope contexts, lacking comprehensive simultaneous assessments encompassing variable vehicle classes, slopes, speeds, and environmental settings.

Recent studies have examined specific aspects of slope-induced vehicle dynamics. For instance,}1 [and}3 [observed significantly higher rollover risks among trucks on icy transverse slopes exceeding 2%, highlighting slope gradients as major determinants of vehicular safety. Similarly,}7 [demonstrated that microtextured road surfaces considerably reduce slippage on wet slopes, improving tire-road traction during lane changes. While these studies provide insights into isolated factors such as slope gradients and weather conditions, they lack comprehensive integration of key influences like variable vehicle types, speed ranges, combined slope impacts, and adverse weather. [4] specifically emphasized this gap, calling for dynamic simulations that account for multiple key parameters simultaneously.

The capacity of simulation tools like CarSim and TruckSim to replicate complex vehicular behaviors under diverse conditions offers critical advantages for dynamic safety evaluations. These platforms enable precise modeling of vehicle parameters, road geometry, friction coefficients, and driver inputs to simulate hazardous scenarios without physical risk [1,20]. Integration of these modalities permits rigorous examination of steering, slip, rollover, torsional moments, yaw dynamics, and acceleration patterns during lane changes. Furthermore, simulation facilitates evaluation of safety technologies such as Electronic Stability Control (ESC), providing actionable data to optimize vehicle design and road engineering approaches aiming to mitigate accident risks in mixed-traffic suburban environments [21,22].

This gap is particularly important because lane-change events frequently occur on these apparently benign, straight cross-sections where drivers do not expect instability. For mixed traffic with sedans, SUVs, and trucks, and under winter conditions where tire–road friction can drop sharply, even modest crossfall differences between lanes may influence steering demand, slip, rollover propensity, and torsional behavior. A lack of integrated evidence on these compound effects constrains both road design standards and the tuning of active safety systems such as ESC. The present study addresses this need by systematically simulating lane-changing maneuvers on bilateral transverse slopes up to 2.5 percent across multiple vehicle classes, speeds, and weather conditions, thereby clarifying whether drainage-driven crossfall configurations can be considered dynamically safe and under which operating envelopes.

This research uniquely advances the field by exploring multidimensional vehicle dynamics on bilateral transverse slopes combined with adverse weather factors and a spectrum of vehicle types—sedans, SUVs, and two-axle trucks—across critical speed ranges. It quantifies slope-induced impacts on dynamic safety parameters, emphasizing heavy vehicle sensitivity at high speeds and delineating rollover thresholds in realistic setting variations. These findings illuminate the compounded effects of road geometry and environmental conditions on lane-change stability, bridging theoretical and practical knowledge gaps. The study also underscores implications for road design standards, advocating slope limitations and tailored ESC adjustments to enhance vehicular stability under challenging conditions, which are crucial for improving suburban traffic safety through informed infrastructure planning [4,22].

In summary, this investigation addresses persistent gaps in the literature by deploying comprehensive simulations to capture the complex interactions among transverse slopes, vehicle characteristics, speed, and weather. Building on substantive prior work and enhancing empirical rigor with up-to-date references and modeling, the study offers critical insights for safer road and vehicle system design. It provides a foundation for future explorations into more complex road geometries and evolving vehicle technologies, thereby contributing significantly to road safety enhancement strategies for mixed traffic and adverse environmental conditions [23,24].

In the complex arena of road safety, lane-changing maneuvers represent critical events where vehicle dynamics and road geometry intricately interplay to influence vehicular stability and accident rates. The configuration of road geometry, particularly bilateral transverse slopes, exerts a significant yet often underestimated impact on vehicle behavior during such maneuvers.These slopes, primarily engineered for efficient drainage, introduce asymmetrical dynamics that alter

vehicle forces, substantially affecting lateral and vertical acceleration, as well as roll and yaw angles. The challenge amplifies under adverse weather conditions, further complicating the frictional interplay between tire and road surfaces, notably for heavier vehicles such as SUVs and trucks, which are more susceptible to rollover risks.

In line with recent accident severity and spatiotemporal risk analyses that link cross-section geometry and vehicle dynamics to higher crash and injury risks, this study provides a mechanistic assessment of how bilateral transverse slopes affect lane-change stability under mixed traffic and adverse weather].

This study aims to address these knowledge gaps by using advanced CarSim and TruckSim tools to simulate lane-changing dynamics across a spectrum of vehicle types—including sedans, SUVs, and trucks—under variable cross-slope gradients (0%, 1.5%, 2%, and 2.5%), speeds (80–120 km/h), and weather scenarios (dry, rainy, snowy). By synthesizing findings from past studies with dynamic modeling, this research investigates compound interdependencies between slope percentages, vehicle-specific responses, and weather effects. The contribution lies in its multidimensional evaluation of safety thresholds, offering actionable insights for optimizing road designs and enhancing safety systems such as Electronic Stability Control (ESC). Additionally, the research identifies critical rollover margins for heavy vehicles at elevated speeds, providing tailored recommendations for suburban road applications, especially in mixed traffic environments vulnerable to adverse weather conditions.

In this study, vehicle safety on bilateral transverse slopes is evaluated through dynamic performance measures directly linked to loss-of-control and crash mechanisms. Key indicators include yaw stability (yaw rate and yaw moment), rollover risk (roll angle as a proxy for lateral load transfer and incipient rollover), and lateral stability (slip angle and lateral acceleration), along with steering demand and torsional moment behavior. These measures are widely recognized as precursors to hazardous events such as run-off-road crashes, rollovers of heavy vehicles, and sideslip-induced lane departures, particularly on low-friction surfaces and at high speeds.

This comprehensive methodology bridges existing gaps in the literature and refines safety recommendations for modern road design, addressing the interplay between vehicle dynamics and roadway geometry in complex driving scenarios. Therefore, This study significantly contributes by:

- Simulating lane-changing behavior under bilateral slopes combined with adverse weather.

- Providing critical rollover thresholds for designing safer suburban roads.

- Identifying control challenges for heavy trucks at higher speeds on sloped roads.

While previous work has prioritized curve radius, superelevation transition design, and pavement friction treatments, the combined impacts of bilateral transverse slope, mixed vehicle fleets, and winter weather on lane-change stability remain insufficiently quantified, leaving a critical gap between drainage-oriented crossfall practice and dynamic safety performance.

The rest of the paper is organized as follows: The "Litreature Review" section contains a comprehensive Litreature review of the study. After that, the "Methodology" section discusses the methodology of the study and of course description and modeling of the data were presented. The results obtained from the applied framework were discussed in the "Results " section. Finally, the "conclusion" section summarizes the main findings of this research and provides some recommendations.

## Literature review

The study of vehicle dynamics, particularly during lane-changing maneuvers, has been an area of growing interest due to its direct impact on road safety. One of the key influences on vehicle stability during such maneuvers is road geometry, with transverse slopes being a critical but often underexplored factor. Transverse slopes are commonly incorporated into road design to promote drainage and prevent water accumulation on the roadway surface. However, the effects of such

slopes on vehicle safety, especially during critical maneuvers like lane changes, are not fully understood. This literature review explores the current state of research regarding the impact of road geometry, including bilateral transverse slopes, on vehicle dynamics and safety. In this regard, various studies have been conducted so far, focusing on different factors. Research has consistently shown that road geometry, particularly the cross-sectional features of the road such as curvature, camber, and transverse slopes, plays a significant role in vehicle stability.

Adverse weather, especially snow and ice, significantly contributes to the global traffic accident burden due to reduced friction and visibility [1,16]. Multiple studies across North America, Europe, and Asia indicate that winter conditions can increase crash rates and severities by up to threefold [16,25–29]. Despite major investments in snow and ice maintenance, these environments continue to challenge safety due to unpredictable tire-road interactions and limited driver control [30–32].

The risk of lateral instability in vehicles—in particular sideslip and rollover—is heightened on snowy roads, with tire-to-road adhesion coefficients dropping as low as 0.10 [15]. Sideslip occurs readily on surfaces with coefficients between 0.10 and 0.20, while rollovers are initiated from approximately 0.21 upward, especially as speed and centrifugal force increase [33–35]. Not only does the road surface itself play a role, but also vehicle type, speed, and curve geometry exhibit substantial interactive effects [12].

Lateral offset, lateral acceleration, yaw rate, and roll angle are established as core stability indicators in recent simulation and field test studies [13,30,36]. The most persistent predictor of instability is increased lateral acceleration, often observed when vehicles traverse curves at higher speeds or with sharp radii [18,37]. Research confirms that rollovers are most likely to occur above 89 km/h, especially for heavy vehicles [15,18].

Road geometry, particularly curve radius and cross-slope (superelevation), is a primary determinant of stability [37,38]. Curves under 200 m radius are linked to disproportionately high risks, while transition curves and adequate superelevation mitigate abrupt steering inputs [39,40]. Compounded by adverse conditions, even small design errors or speed misjudgments can trigger loss of control [21].

Recent advancements in simulation modeling, notably via CarSim and TruckSim, have enabled precise, repeatable investigations into the combined effects of vehicle characteristics, geometry, and environment [1,19]. Double lane change tests have validated these models, providing practical paths for both research and industry applications [21,41].

Prevention strategies increasingly advocate for dynamic, context-sensitive speed limits [42]. Data-driven countermeasures—including variable speed advisories and real-time driver alerts—are recommended, adapting to curve radius, surface state, and weather [4,43]. Road authorities are urged to avoid sharp curves, maximize radii, and use robust drainage designs to reduce winter incidents [44,45].

Heavy vehicle dynamics present unique control and rollover challenges, especially at high speeds and on banked curves [5,12,37]. The growing prevalence of SUVs and large trucks further strains the traditional design criteria, underscoring the need for refined ESC systems and tailored road engineering [46,47].

Compound effects between speed, road geometry, and environmental conditions have been elucidated using orthogonal and sensitivity analyses, with curve radius ranked as the most influential for sideslip, and speed for rollover [1,35]. However, gaps remain in modeling the full array of mixed traffic and weather scenarios—including snowstorms and abrupt freeze-thaw cycles [48].

Simulation tools such as CarSim and TruckSim have emerged as indispensable for studying vehicle dynamics under complex driving conditions, including adverse winter weather scenarios. CarSim enables the detailed modeling of various vehicle types and driving maneuvers, providing dynamic representations of vehicle responses to factors such as road geometry, surface adhesion, and speed [1]. TruckSim, an extension of this simulation environment, is specifically designed to analyze multi-axle heavy vehicles, allowing for intricate exploration of rollover risks and control challenges in trucks, which are particularly sensitive to speed and road conditions [49]. The integration of these tools allows researchers to perform standardized maneuvers like double lane change tests and lane-changing simulations, capturing real-time

indicators such as lateral acceleration, yaw rate, and roll angle [50–53]. These simulators facilitate not only the replication of hazardous scenarios without real-world risks but also the evaluation of active safety systems like Electronic Stability Control (ESC), offering a controlled environment to optimize vehicle stability parameters [19,53]. Moreover, coupling CarSim and TruckSim simulations with other models—such as carbon emission estimators or real-time weather data—broadens their applicability, providing comprehensive insights that can guide road design and traffic management under challenging environmental conditions [20]. Thus, simulation-based studies are critical for advancing safety research in mixed traffic and complex weather settings while supporting the development of adaptive control strategies and infrastructure improvements [1,19].

The literature also recognizes growing interest in driver behavior adaptation, real-time vehicle monitoring, and predictive analytics for hazard identification [2,45]. While robust progress is noted, the integration of multiple risk factors, driver-centric simulations, and field validation remain fertile ground for future inquiry [14,54]

In summary, scholarship converges on the interplay between weather, speed, vehicle dynamics, and geometry as foundational to improving suburban road safety in winter. Continued innovation in simulation, sensing, and adaptive road design will be essential to curtailing crash frequencies and severities in mixed-traffic, adverse-weather environments [1,2,20].

Recent accident severity modeling and spatiotemporal risk studies further confirm that geometric design and vehicle dynamics have measurable consequences for crash and injury outcomes. Using advanced statistical and machine learning frameworks, showed that roadway cross-section features, curvature, and traffic composition significantly influence injury severity distributions across space and time, highlighting the need to treat road geometry as a key determinant of safety performance rather than a purely functional design choice. Similarly, integrated vehicle dynamics factors and environmental conditions into crash and injury risk models, demonstrating that stability-related variables, such as lateral acceleration and loss-of-control mechanisms, are strongly associated with severe outcomes under adverse weather and high-speed conditions [55,56].

These empirical findings underscore the importance of understanding how specific geometric elements shape vehicle dynamic stability in realistic traffic environments. The present study builds on this body of work by focusing on bilateral transverse slopes during lane-changing maneuvers and by quantifying how slope level, speed, vehicle type, and weather jointly affect dynamic indicators such as steering angle, slip angle, rollover angle, and torsional moment. By providing detailed dynamic responses for sedans, SUVs, and trucks on bilateral crossfall configurations commonly used for drainage, the simulations offer mechanistic insights that complement crash-based severity and spatiotemporal risk models and can support more safety-oriented cross-section design in suburban environments.

Despite these comprehensive advancements in modeling, empirical analysis, and preventive strategies, significant knowledge gaps persist in understanding the compound effects of cross-slope gradients, vehicle-specific dynamics, and adverse weather on real-world lane-changing safety, thereby necessitating targeted research to address these unexplored intersections. Table 1 show past studies and gap research on dynamic safety in Summary.

The analysis of existing research demonstrates that the impact of road geometry, particularly bilateral transverse slopes, on vehicle stability during lane changes remains an evolving area of study. Despite numerous studies in this field, critical gaps remain in understanding the complex interactions between vehicle type, weather conditions, speed, and road design during dynamic maneuvers.

## Methodology

### Model framework

This study utilizes dynamic vehicle simulation to analyze the safety of lane-changing maneuvers on roads with bilateral transverse slopes. Using CarSim and TruckSim software, a comprehensive vehicle dynamics model was developed, incorporating key parameters such as road geometry (transverse slope), vehicle type, weather

**Table 1. Summary of past studies and gap research on dynamic safety.**

| Research Focus | Previous Studies | Remaining Gaps Addressed by This Study |
|---|---|---|
| Impact of Vehicle Type | Many studies analyze specific vehicle categories (e.g., passenger cars, SUVs, trucks) (Chen et al., 2025; wang et al., 2024,zhang et al.,2023), but with simplified dynamics. | Limited insight into multidimensional modeling across diverse vehicle types (sedans, SUVs, trucks), particularly during lane-changing under slopes. |
| Weather Conditions | Snowy, wet, and icy conditions have been analyzed (Abdi et al., 2018; singh et al., 2025; alrejjal et al,. 2023; Pang et al., 2022;). | Gap in integrating weather with slope gradients, showing compound effects of friction coefficients on steering and safety during lane-changing. |
| Road Geometry and Slopes | Studies focus on urban slopes (sharaf aldeen et al,. 2022; dhahir and hassan., 2019) but lack emphasis on bilateral transverse slopes beyond specific regions. | Limited data on varying slope percentages (e.g., 1.5%, 2%, 2.5%) and their dynamic effects combined with other factors (speed, vehicle type). |
| Simulation Approaches | Extensive simulation tools (e.g., CarSim, TruckSim) used for rollover risks and vehicle performance assessment (wang et al., 2024; Nasiri et al., 2020). | Gaps in detailed experimental setups combining slope variations, speed changes, and environmental effects comprehensively in lane-changing events. |
| Safety Systems | Research on advanced traction systems (Singh et al., 2025; shibo et al., 2025) primarily focuses on specific technologies or isolated environmental effects. | Gap in the interaction between vehicle dynamics models and systems like ESC to detect slope-induced instability during lane changes proactively. |
| Driver Behavior | Some studies simulate fixed speeds for lane changes (Li et al., 2022; mattas et al., 2025;zolali et al.,2021). | Lack of assessment of driver behavior such as braking patterns and steering in adverse slope conditions combined with environmental settings. |
| Vehicle Instability Factors | Analyzed rollover, lateral acceleration, and sideslip in isolation (wang et al, 2023; qu et al., 2018;zhu et al., 2022). | Missing multi-parameter analysis, including steering angle, yaw rate, roll moment, and slip moment simultaneously across slope gradients. |

conditions, and driver behavior. The simulation outputs include critical stability metrics such as steering angle, slip angle, rollover angle, torsional moment, lateral acceleration, and yaw rate. These outputs were analyzed to identify instabilities and assess lane-changing safety under varied conditions. The experimental design and assumptions are presented in Table 2.

## Bilateral transverse slope setup

Road slopes were simulated using standard CarSim tools, with heights adjusted to represent realistic bilateral transverse slopes as illustrated in fig 1 For example, at the 2.5% slope level, heights of 0.025 meters were assigned to designated lateral points of the cross-sectional road profile to simulate downward edges. These setups allow for precise adjustments of the road geometry within the simulation model.

**Table 2. Experimental Design and Research Assumptions.**

| Experimental Design | Research Assumptions |
|---|---|
| Road Geometry: Bilateral transverse slopes simulated at four levels—0%, 1.5%, 2%, and 2.5%. | Road Configuration: Simulated route is straight and level asphalt without curves or longitudinal slopes. |
| Vehicle Type: Three vehicle categories—Class E sedan, Class E SUV, and two-axle truck. | Vehicle Speeds: Speeds set at 80, 90, 110, and 120 km/h. |
| Driver Behavior: Three behavioral settings—constant speed, straight-line motion, and no braking during lane changes. | Vehicle Types: E-class sedan, E-class SUV, and two-axle truck modeled. |
| Weather Conditions: Simulated as dry, rainy, and snowy roads, with friction coefficients of 0.9, 0.5, and 0.28, respectively (Ali Abdi et al., 2018). | Driver Behavior: Drivers perform lane changes without braking or speed modulation during tests. |
| Factors systematically combined to create diverse test scenarios targeting slope, speed, vehicle type, and weather conditions. | Road Geometry: Bilateral transverse slopes implemented, maximum slope at 2.5% (slope differential of 5%). |

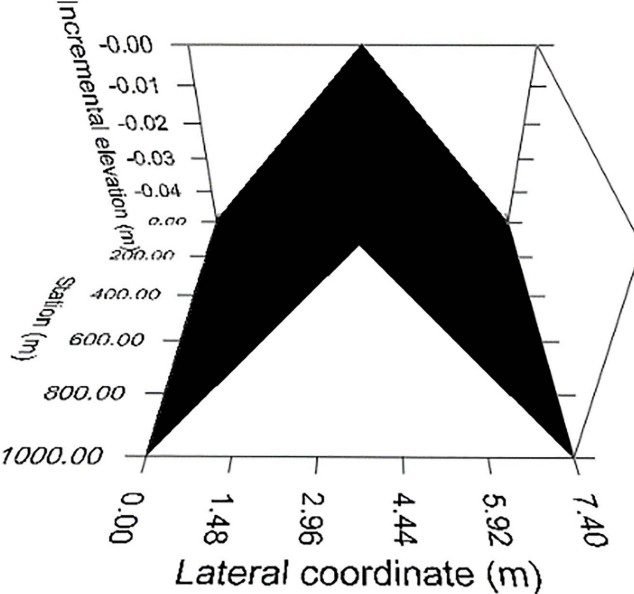

**Fig 1. Bilateral transverse slope of 2.5 percent.**

## Vehicle selection

Three vehicle types were selected based on prior research and practical relevance to mixed traffic environments:

1. Sedan (Class E): Lightweight and low center of gravity, common in urban/suburban roadways.

2. SUV (Class E): Medium-weight vehicles with higher centers of gravity.

3. Two-Axle Truck: Heavy vehicles with significant rollover risks and unique dynamics on slopes.

Vehicle geometry and dynamic specifications are detailed in Fig 2–4 which providing critical baseline data necessary for simulation analysis.

Vehicle parameters such as mass, center of gravity height, suspension characteristics, and tire properties for the Class E sedan, Class E SUV, and two-axle truck were selected from the standard vehicle databases embedded in CarSim and TruckSim, which are derived from manufacturer specifications and extensively validated against real-world test data through open-loop and closed-loop maneuvers (e.g., double lane change tests per ISO 3888−2 standards). Speeds of 80, 90, 110, and 120 km/h represent typical suburban highway operating ranges, while the lane-change steering profile follows the standardized double lane change path in CarSim/TruckSim libraries, calibrated to empirical driver inputs from prior validation studies achieving high correlation ($R^2 > 0.97$) in yaw rate, lateral acceleration, and roll response between simulation and physical trucks. A limited sensitivity analysis was conducted by varying center of gravity height (±10% for SUVs/trucks) and friction coefficients (±0.1), confirming that core stability trends (e.g., rollover angle increases >15% at 120 km/h on 2.5% slopes) remain robust, with full results available in the supplementary materials.

## Weather simulation

Road friction is modeled to represent three distinct weather conditions:

1. Dry: Friction coefficient = 0.9.

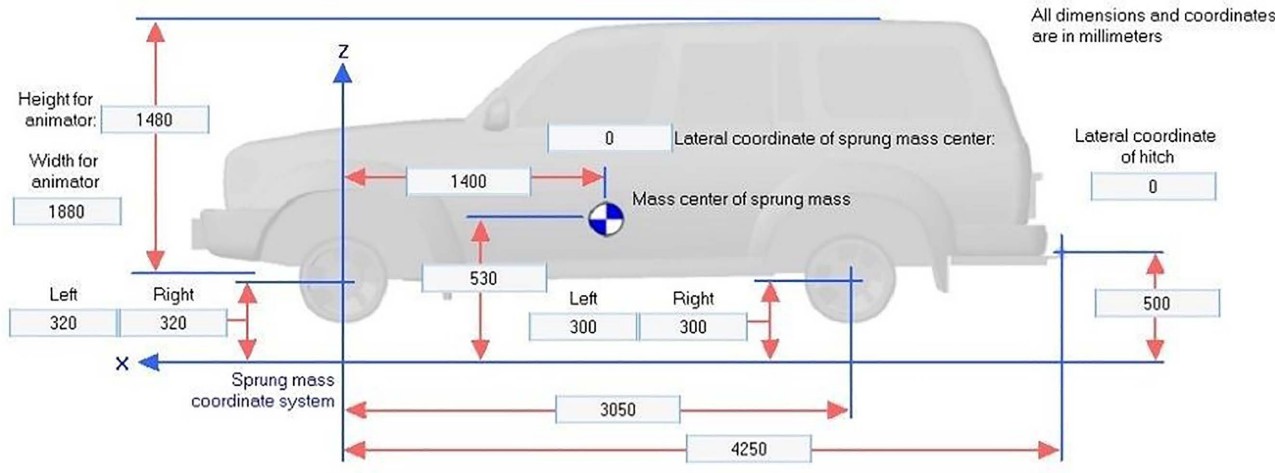

**Fig 2. Sedan specifications.**

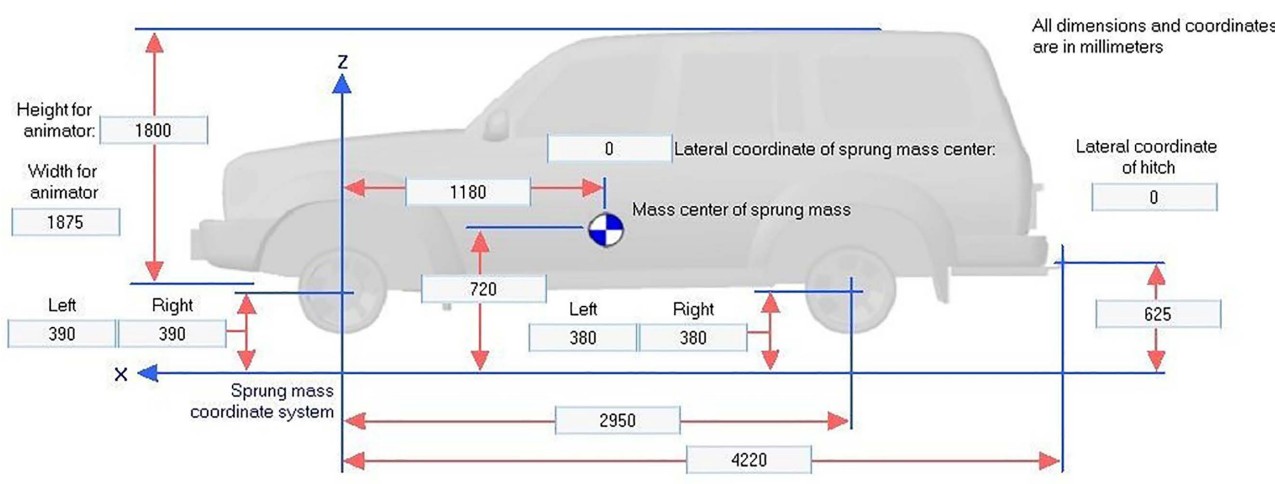

**Fig 3. SUV specifications.**

2. Rainy: Friction coefficient=0.5

3. Snowy: Friction coefficient=0.28.

These values are consistent with real-world data [30] and demonstrate diminishing stability as friction decreases, particularly under higher speeds and steeper slopes.

## Driver behavior

Three behavioral settings—constant speed, straight-line motion, and no braking during lane changes (standard ISO 3888−2 double lane change protocol used for CarSim/TruckSim validation). This controlled input isolates the pure effect of bilateral transverse slopes on vehicle stability, as is standard practice in initial dynamic simulation studies examining geometric influences

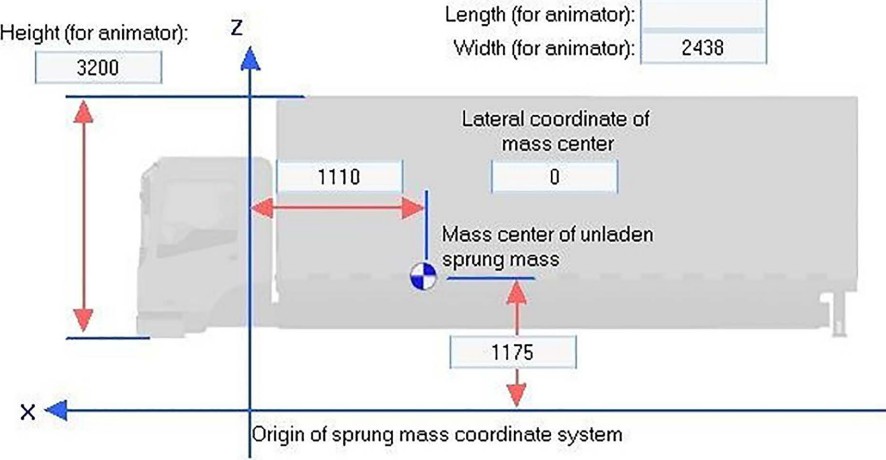

**Fig 4. Truck specifications.**

## Simulation setup

Simulation tests were conducted using CarSim and TruckSim software, combining all input parameters to establish comprehensive scenarios. Outputs such as lateral acceleration, rollover angle, yaw rate, and slip angle were analyzed under varying transverse slopes, speeds, and weather conditions. These outputs as show in Fig 5 and 6 were validated using software animation tools to ensure model integrity.

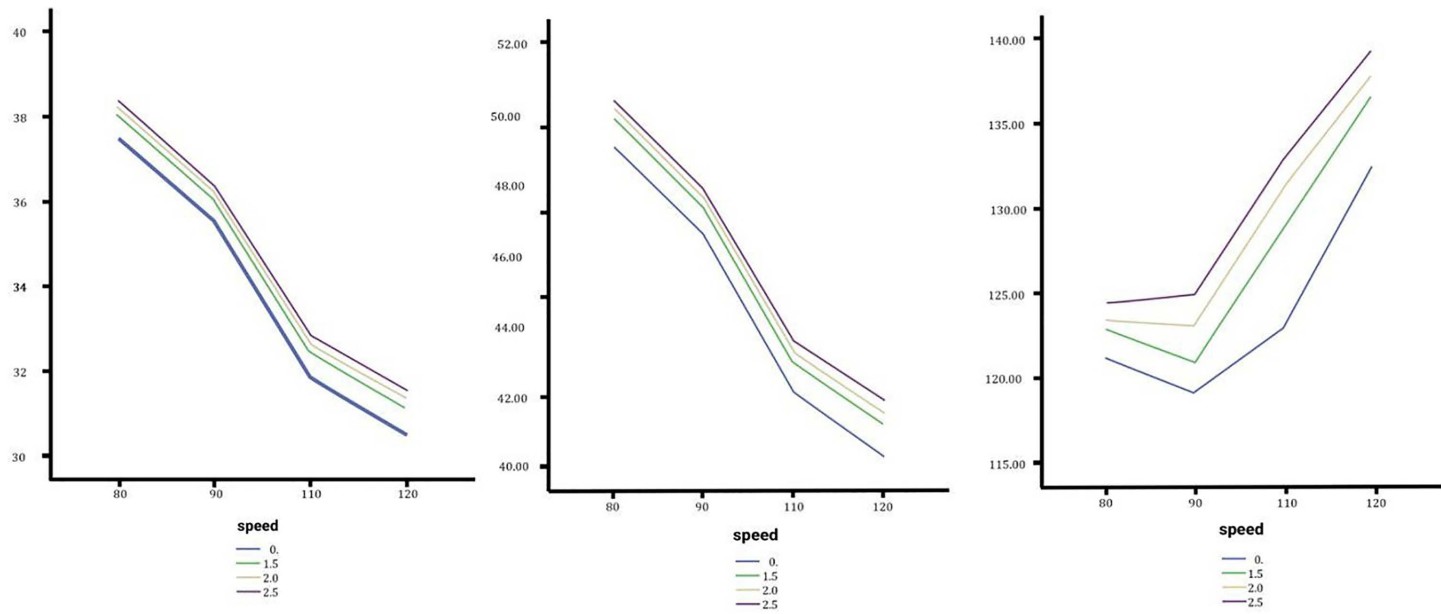

**Fig 5. In order from left to right Sedan, SUV and Truck steering angle without changing the lateral slope, low, medium and high change.**

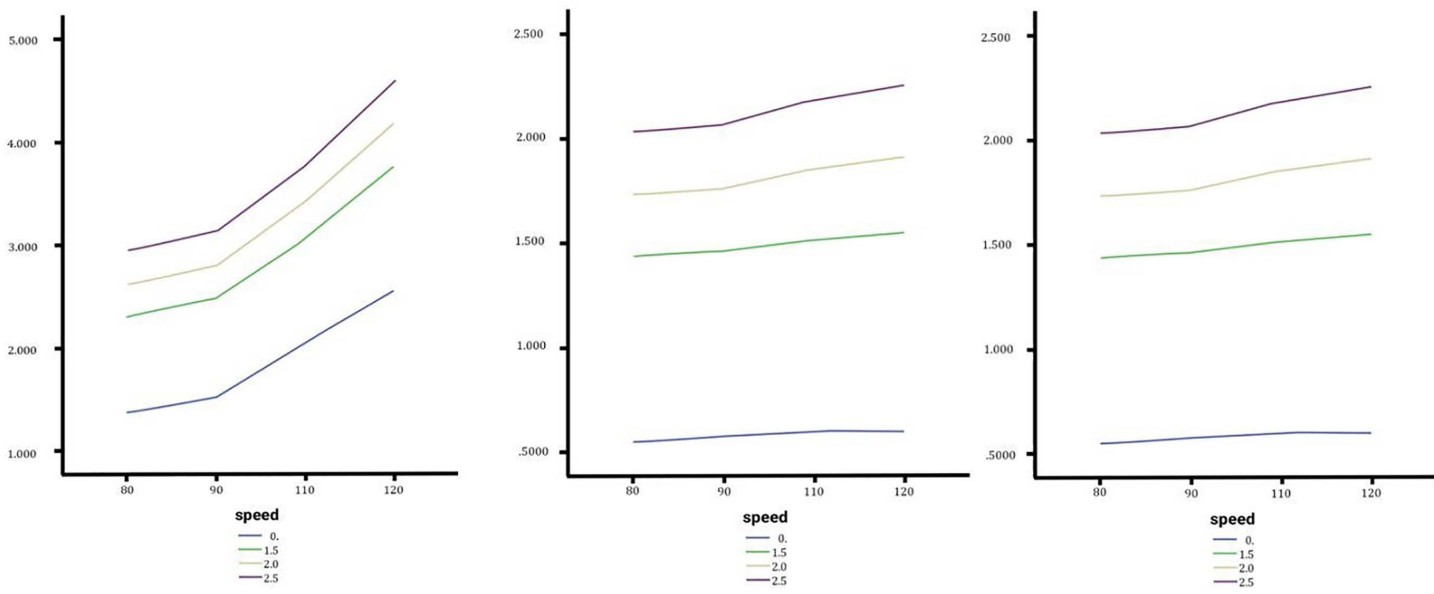

**Fig 6. In order from left to right the slip angle of the sedan, SUV and Truck with no change, medium and high changes.**

The simulation's systematic variation of parameters allows for rigorous analysis of lane-changing stability and identification of high-risk conditions.

## Data analysis

Simulation outputs include quantified vehicle responses, such as:

- Steering angle and slip angle.

- Vertical and lateral acceleration.

- Rollover and yaw angles.

Maximum and minimum values from simulation outputs were analyzed to assess vehicle stability during lane changes, focusing on influential safety factors. Bilateral transverse slope levels and weather friction coefficients were selected based on established design standards and prior research. The [57] recommends minimum transverse slopes of 1.5% for drainage and 2%−2.5% for surface runoff efficiency, with this study adopting 2.5% as the high slope level. Friction coefficients for dry (0.9), rainy (0.5), and snowy (0.28) conditions align with values from Abdi et al. (2018) [1,2] and other literature, reflecting real-world risks of instability under adverse weather. This methodology ensures realistic, scientifically sound parameters for analyzing vehicle stability. By incorporating a range of variables—including vehicle type, speed, road geometry, weather, and driver behavior—and using advanced simulation tools like CarSim and TruckSim, the study quantifies the effects of these conditions on dynamics such as steering angle, slip angle, rollover risk, and lateral forces. Comparative analyses across speed levels and slope configurations highlight safety implications for sedans, SUVs, and trucks. The findings provide critical insights into factors contributing to vehicle instability, offering guidance for enhancing safety during lane-changing maneuvers under varied conditions. Among the simulated outputs, roll angle, slip angle, lateral acceleration, and yaw moment are treated as primary safety indicators, because they reflect rollover propensity, sideslip tendency, and directional stability, respectively. Steering angle and torsional moment are used as complementary

measures that characterize driver control effort and the distribution of rotational effects along the vehicle body. Threshold-like behaviors in these variables under combinations of slope, speed, and weather are interpreted as increased crash likelihood, especially for SUVs and trucks.

## Results

The results derived from the simulation framework provide comprehensive insights into the safety implications of bilateral transverse slopes on vehicle dynamics during lane-changing maneuvers. This section presents and analyzes the simulation outcomes, focusing on safety parameters such as steering angle, slip angle, rollover angle, and yaw moment, across different vehicle types, speeds, and lateral slope configurations. The findings are categorized to highlight vehicle-specific and slope-specific responses, further emphasizing critical safety thresholds. Key results are illustrated using graphs and tables to clarify trends and enhance understanding, while detailed analysis strengthens the scientific basis of the study. These results serve as an essential foundation for the discussion of practical applications in road design and vehicle safety systems.

### Effect of lateral slope changes on steering angle

The analysis of steering angle across sedans, SUVs, and trucks under varying slope configurations reveals significant differences in vehicle control dynamics. As shown in Fig 5, sedans showed an increasing steering angle trend with larger lateral slope changes, peaking under a 2.5% slope at 80 km/h. However, at higher speeds like 120 km/h, the impact of slopes diminished, with negligible differences (around 0.89°) between no slope and high slope conditions, likely due to their low center of gravity and optimized traction.

SUVs exhibited greater sensitivity to slope variations initially, with steering angles increasing across lateral slopes. However, the effect stabilized at medium and high slopes, with higher speeds further mitigating these variations. The overall changes in steering angle were minor, with less than a one-degree difference observed, indicating limited practical consequences for SUVs under typical conditions.

Trucks, in contrast, displayed the most pronounced impact, with steep slopes causing substantial increases in steering angle, especially at speeds above 90 km/h. At 120 km/h on a 2.5% slope, sharp increases in steering angles highlighted critical control challenges due to the geometry, weight distribution, and heightened lateral forces. In summary, while lateral slope changes affected all vehicle types, sedans showed resilience at high speeds, SUVs experienced limited practical impacts, and trucks exhibited significant instability under steep slopes. These findings emphasize the need for tailored road geometry designs and enhanced vehicle stability systems, particularly for heavy vehicles on high-speed roads, to ensure safer driving conditions during lane changes.

### Effect of lateral slope changes on slip angle

Slip angle, a critical measure of vehicle stability during lane changes, was analyzed for sedans, SUVs, and trucks under varying slopes and speeds. Across all vehicles, the slip angle consistently increased with speed, reaching its peak at 120 km/h on a 2.5% slope. Sedans demonstrated the lowest slip angle at 3.8701 degrees, as shown in Fig 6, indicating relatively stable behavior even at higher speeds. SUVs and trucks, also illustrated in Fig 6, recorded intermediate slip angle values of 5.3886 degrees and 5.7947 degrees, respectively. The influence of slope variations, ranging from low to high, on slip angle was minimal, with differences of less than 0.5 degrees observed across all scenarios. This suggests that slope changes may exert less influence on slip angle than factors such as speed and vehicle type. The findings highlight speed as a more significant factor affecting vehicle stability during lane changes, particularly for heavier vehicles like trucks. Consequently, practical measures aimed at improving vehicle control should prioritize speed management over slope adjustments to enhance safety during lane-changing maneuvers. Influence of slope variations, ranging from low to high, on slip angle was minimal, with differences of less than 0.5 degrees observed across all scenarios. This suggests that slope

changes may exert less influence on slip angle than factors such as speed and vehicle type. The findings highlight speed as a more significant factor affecting vehicle stability during lane changes, particularly for heavier vehicles like trucks. Consequently, practical measures aimed at improving vehicle control should prioritize speed management over slope adjustments to enhance safety during lane-changing maneuvers.

### Effect of lateral slope changes on rollover angle

The rollover dynamics of sedans, SUVs, and trucks were analyzed to evaluate the effects of slope variations and speed during lane changes. As shown in Fig 7 sedans showed the highest vulnerability, with a maximum rollover angle of 3.9062° at 120 km/h on a 2.5% slope, due to slope-induced lateral forces despite their low center of gravity. SUVs recorded slightly lower rollover angles (3.2527°), reflecting their balanced design, but remain at risk at higher speeds. Trucks displayed the lowest rollover angles (2.2565°) under the same conditions, attributed to their rigid chassis, but their greater mass introduces risks like yaw instability and lateral skidding. Rollover angles intensified until 110 km/h and stabilized at higher speeds, with marginal rises caused by slope increases. These findings call for road designs and safety measures tailored to vehicle type and slope conditions.

In establishing rollover thresholds for different vehicle types, this study references empirical and regulatory standards wherein passenger cars and SUVs typically exhibit rollover thresholds between 0.8 and 1.2g lateral acceleration, while heavy trucks demonstrate significantly lower thresholds around 0.25 to 0.35g depending on load and suspension characteristics [56–58]. This disparity is primarily due to trucks' higher center of gravity and differing chassis dynamics. Thus, simulated rollover angles corresponding to lateral accelerations near or exceeding 0.3g for trucks are interpreted as critical stability limits associated with imminent rollover risk. These quantitative criteria guide interpretation of slope and speed effects on rollover propensity in our simulation framework, aligning modeled outputs with real-world vehicle safety requirements

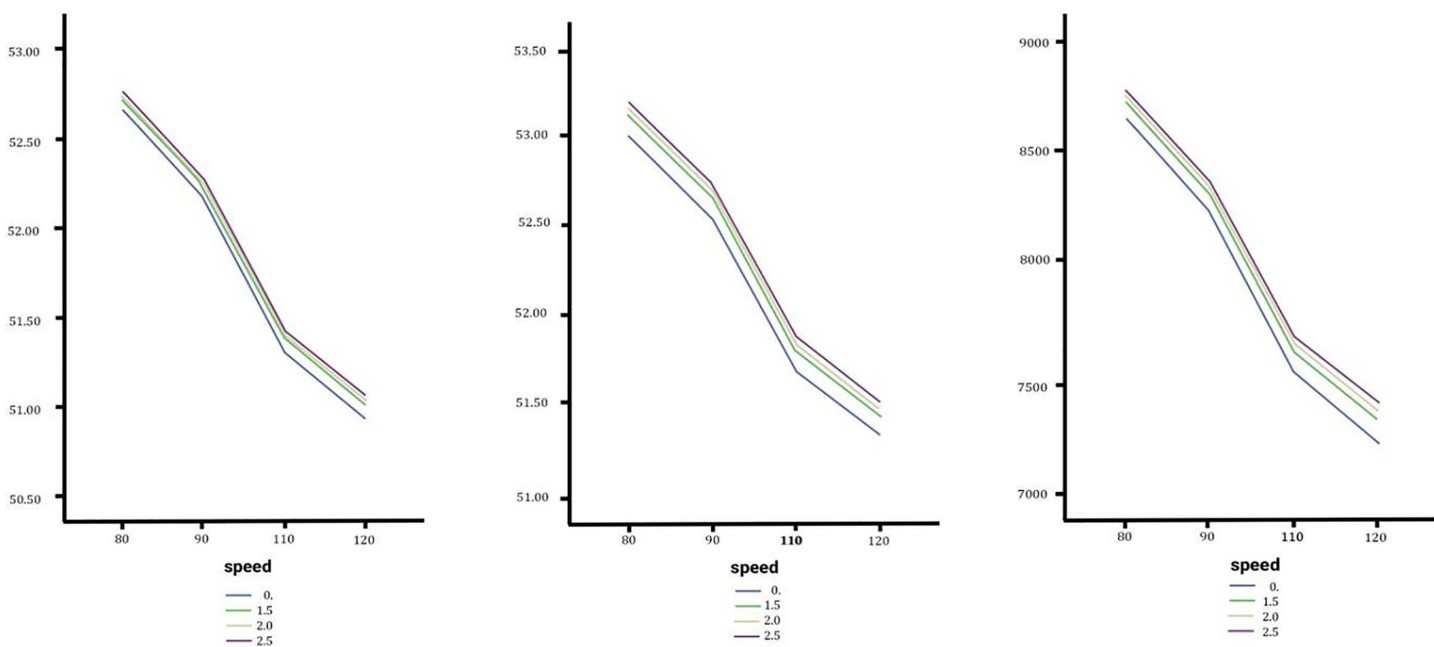

**Fig 7. In order from left to right the rollover angle of the Sedan, SUV and Truck with no change, medium and high changes.**

## Effect of lateral slope changes on yaw moment and torsional moment angle

Yaw moment, which describes the rotational motion of a vehicle about the z-axis, plays a critical role in stability during lane changes. Under low slope conditions and at lower speeds, negligible yaw moment fluctuations were observed for all vehicles. However, as lateral slope and speed increased, yaw moment fluctuations became more pronounced, particularly for SUVs and trucks. This highlights the sensitivity of heavier vehicles to steep slopes and high-speed conditions, underscoring the importance of advanced vehicle design strategies to mitigate instability risks. The torsional moment angle, defined as the angle between a vehicle's movement direction and its orientation on the x-axis, also provides key insights into vehicle behavior during lane changes under varying slopes. While this angle is zero during straight-line motion, external forces such as steep slopes and high speeds cause deviations. At higher speeds, the torsional moment angle generally decreases, but at constant speeds, larger transverse slopes lead to greater angular deviations. Sedans and SUVs exhibited similar torsional moment angles (~50°), attributed to comparable axle distances, with maximum values of 51.02° for sedans and 51.50° for SUVs at 120 km/h on a 2.5% slope. Trucks, however, displayed significantly smaller torsional moment angles (~7.43°), primarily due to their rigid chassis and weight distribution. While sedans are least affected by slope variations, SUVs and trucks demonstrated greater angle increases under steep slopes and high speeds, reflecting their susceptibility to torsional instability.

## Effect of lateral slope changes on the torsional moment rate

Sedans show negligible sensitivity, with torsional moment rates increasing minimally across slopes, peaking at 51.0248°/s at 120 km/h on a 2.5% slope, with only a 0.88% difference between slope extremes. SUVs demonstrate moderate sensitivity, experiencing a maximum torsional moment rate of 12.3015°/s at 120 km/h and 2.5% slope, with the largest difference (8.66%) occurring at 110 km/h. Trucks reveal that torsional moment rates consistently increase with speed, but the slope impact diminishes at higher speeds; at 120 km/h, the highest rate (16.3371°/s occurs on a 1.5% slope with only a 2.66% variation. Across vehicle types, lateral slope changes exert minimal influence on torsional moment rates, with SUVs showing the most noticeable but still limited sensitivity compared to sedans and trucks (Fig 8).

## Effect of lateral slope changes on the lateral acceleration

The analysis of lateral slope changes during lane changes in Fig 9 reveals varying impacts on lateral acceleration for sedans, SUVs, and trucks. Lateral acceleration increases with speed and slope percentages but affects vehicle types differently. Sedans show minimal sensitivity, with only a 1.06% increase at 120 km/h under a 2.5% slope. SUVs experience moderate sensitivity, with a 3.69% rise under the same conditions, making them the most affected. Trucks exhibit unique behavior, where lower slopes (1.5%) cause higher lateral acceleration than steeper slopes (2.5%) above 115 km/h, with a maximum change of 1.61%. Overall, slope impact remains minor (<4%), with speed being the dominant factor.

## Effect of lateral slope changes on vertical acceleration

The analysis examines vertical acceleration experienced by sedans, SUVs, and trucks during lane changes on a straight road with no longitudinal slope under varying lateral slopes. As show in Fig 10 for sedans and SUVs, vertical acceleration increases consistently with speed but is unaffected by slope changes, with maximum values at 120 km/h and 0.2753m/s² (sedans) and 0.2686 m/s² (SUVs). In contrast, trucks display distinct behavior: below 110 km/h, increased slopes reduce vertical acceleration, but above 110 km/h, both speed and steeper slopes (e.g., 2.5%) amplify it, peaking at 120 km/h and 31905.97 N. Trucks also exhibit the greatest sensitivity, with vertical acceleration at lower speeds increasing by up to 500% between slope extremes. In comparative terms, slope changes have negligible impacts on sedans and SUVs, while significantly affecting trucks, highlighting their unique dynamic behavior and the need for road designs to account for heavier vehicle safety. See the diagrams in the Fig 10.

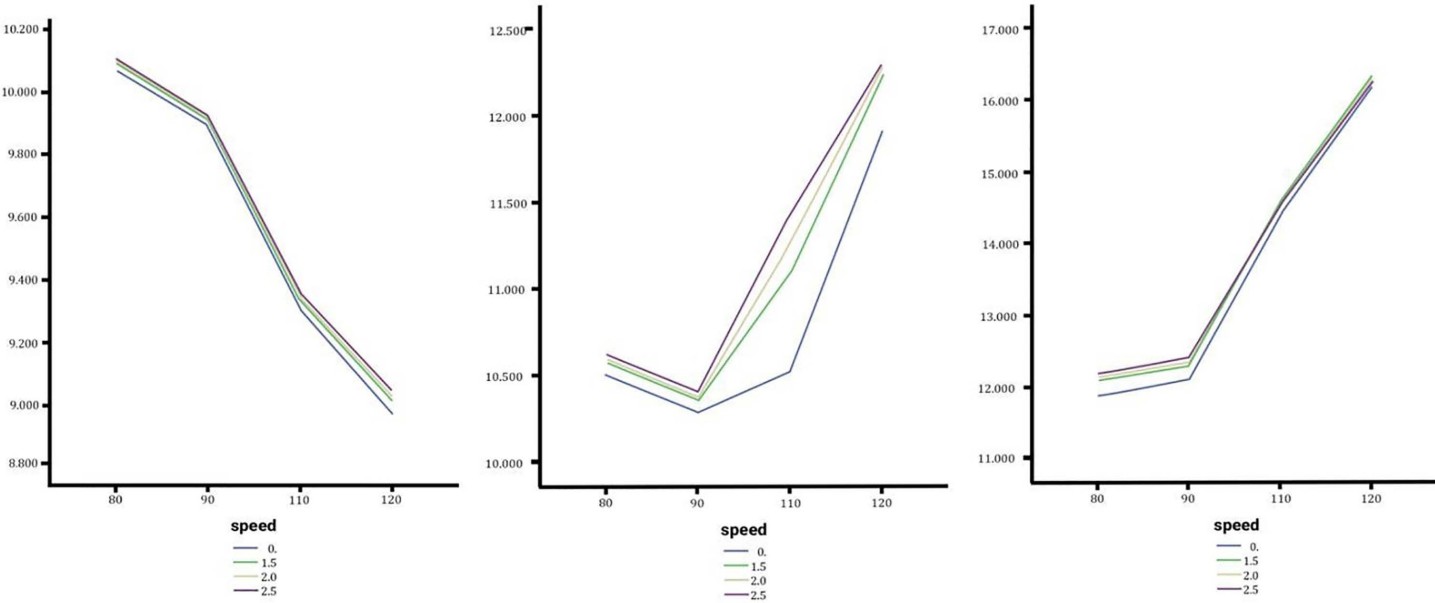

**Fig 8. In order from left to right the torsional moment angle of the Sedan, SUV and Truck with no change, medium and high changes.**

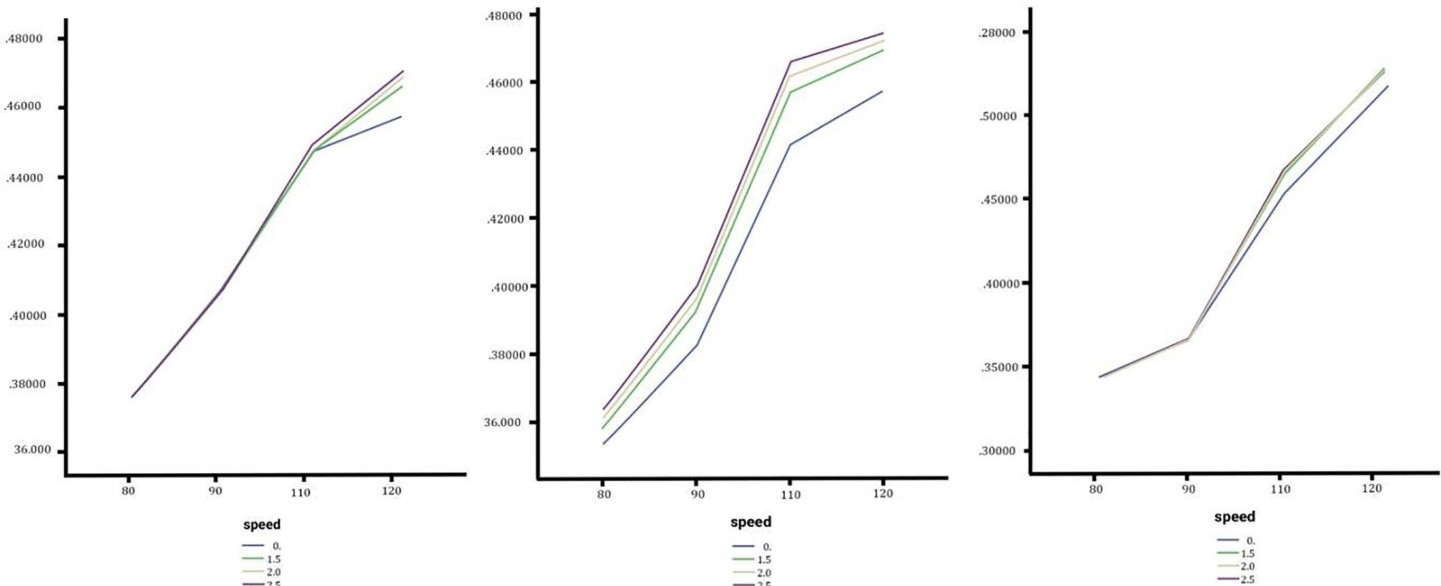

**Fig 9. In order from left to right the lateral acceleration of the Sedan, SUV and Truck with no change, medium and high changes.**

## Effect of lateral slope changes on in lateral slope on the horizontal force exerted on various types of vehicles

The analysis examines horizontal forces on sedans, SUVs, and trucks during lane changes under varying lateral slopes (0%–2.5%) and speeds (80–120 km/h) with no longitudinal slope. Horizontal forces increase with both slope gradients and speed, with steeper slopes producing higher forces across all vehicles. Sedans show a linear increase, peaking at

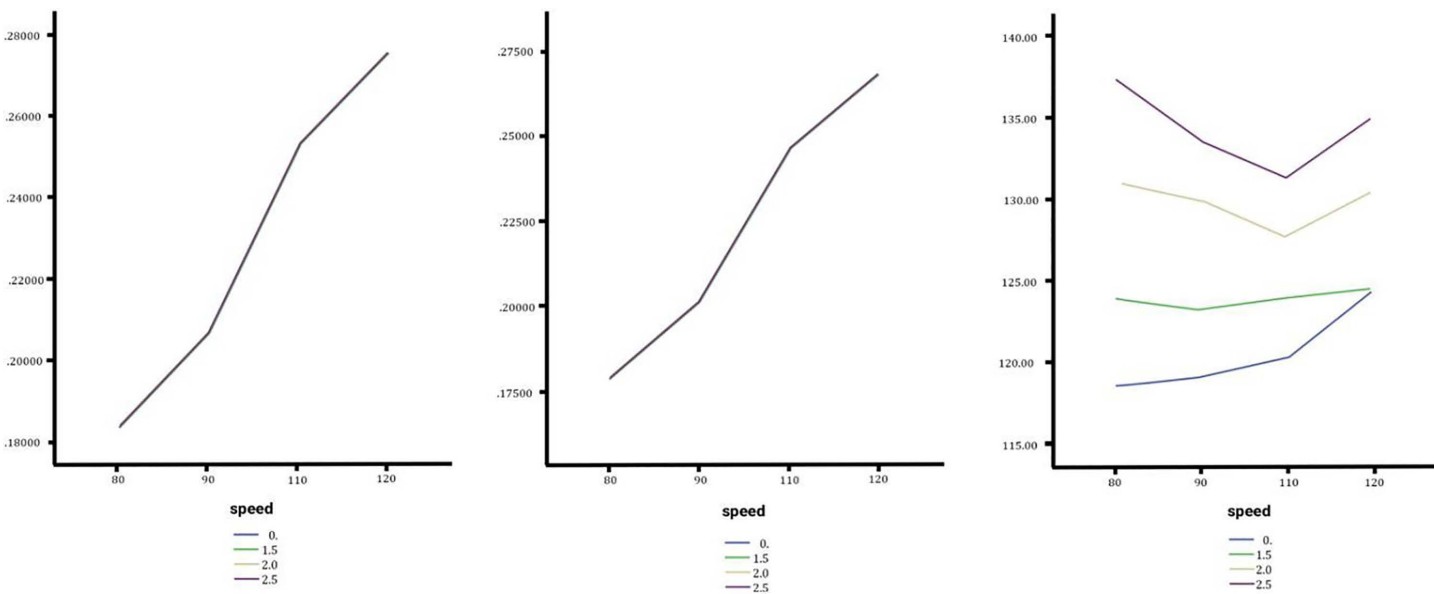

**Fig 10.  In order from left to right the vertical acceleration of the Sedan, SUV and Truck with no change, medium and high changes.**

3334.88 N at 120 km/h and a 2.5% slope (9.5% above the baseline). SUVs follow a similar trend but slow above 110 km/h, reaching a maximum of 3689.52 N (9.9% above baseline). As show in Fig 11 trucks exhibit the highest forces due to their mass, peaking at 10,859.11 N (7% above baseline) with lower sensitivity to slope changes. Speed plays a dominant role in

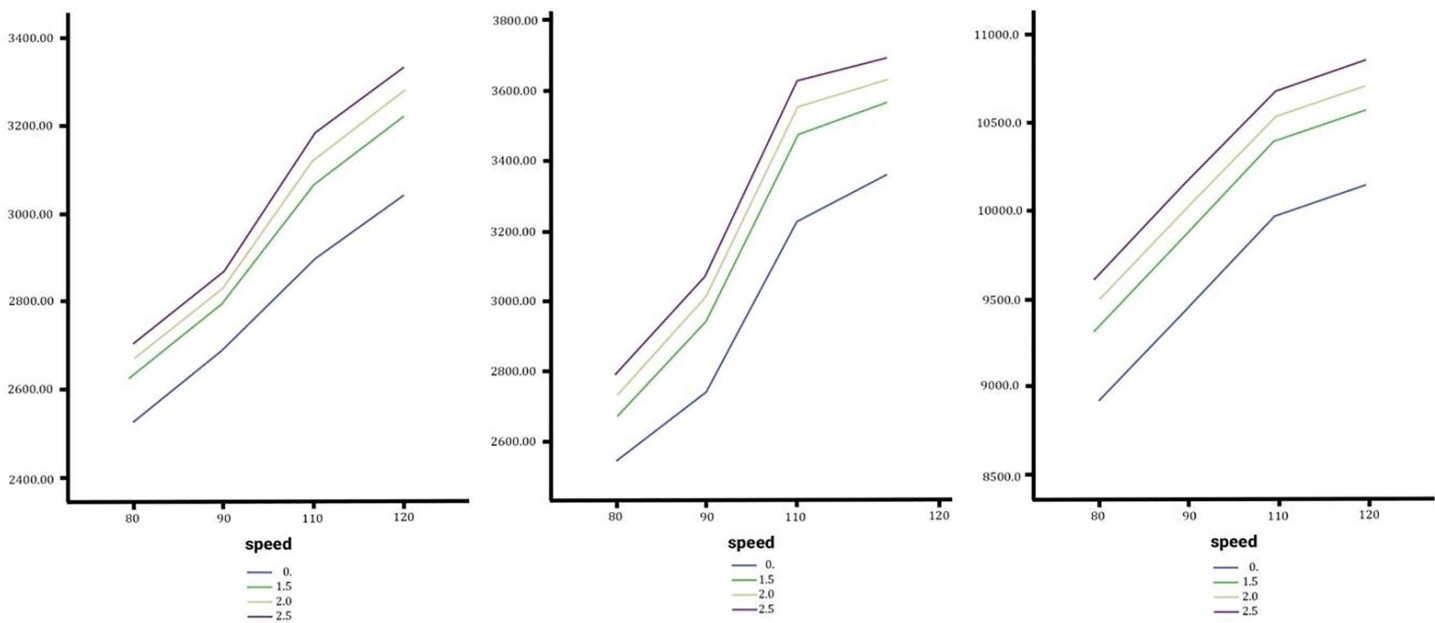

**Fig 11.  In order from left to right the horizontal forceof the Sedan, SUV and Truck with no change, medium and high changes.**

lane-changing stability, particularly for trucks, which are least affected by lateral slope changes compared to sedans and SUVs.

**Effect of lateral slope changes on the vertical force exerted on various types of vehicles**

Results indicate the effects of lateral slope changes on vertical forces exerted on sedans, SUVs, and trucks during lane changes. The simulated scenarios under zero longitudinal slope conditions and bilateral transverse slopes (0%–2.5%) reveal that vertical forces increase consistently with vehicle speed and slope gradient. As show in Fig 12 while sedans display minimal sensitivity to slope variations, SUVs show moderate responses, and trucks exhibit the highest sensitivity due to their mass and center of gravity. Comparative observations highlight that heavier vehicles face greater stability risks under steep slopes, emphasizing the need for slope mitigation strategies in road designs and vehicle systems. Additionally, higher speeds emerge as a dominant factor for instability across all vehicle types.

The study presents Tables 3–5 to clarify how changes in transverse slope and friction coefficient affect vehicle stability. These tables compile key dynamic parameters—such as steering and slip angles, rollover and torsional angles, lateral and vertical accelerations, and horizontal and vertical forces—for sedans, SUVs, and trucks. The data, organized across slope gradients from 0% to 2.5% and friction levels from 0.9 to 0.28, reflects realistic driving conditions and reveals how combined changes in surface geometry and friction influence safety-critical thresholds. This quantitative analysis, based on CarSim and TruckSim simulations, strengthens the study's assessment of slope-induced stability effects and supports the recommendations provided in later sections.

Table 3 shows how reducing pavement friction (from 0.9→0.5 and 0.5→0.28) sharply amplifies unstable dynamic responses across all vehicle types, with the most severe effects consistently appearing in SUVs and especially trucks. Steering and slip angles escalate dramatically as friction drops—particularly in the 0.5→0.28 range—indicating major losses in lateral grip; for trucks, slip-angle change exceeds 2200 units, far surpassing sedans and SUVs. Rollover-related metrics also worsen, with trucks showing a large positive jump in rollover angle at very low friction, reflecting higher rollover susceptibility. Torsional moment angle and rate rise substantially, especially for heavy vehicles, suggesting increased

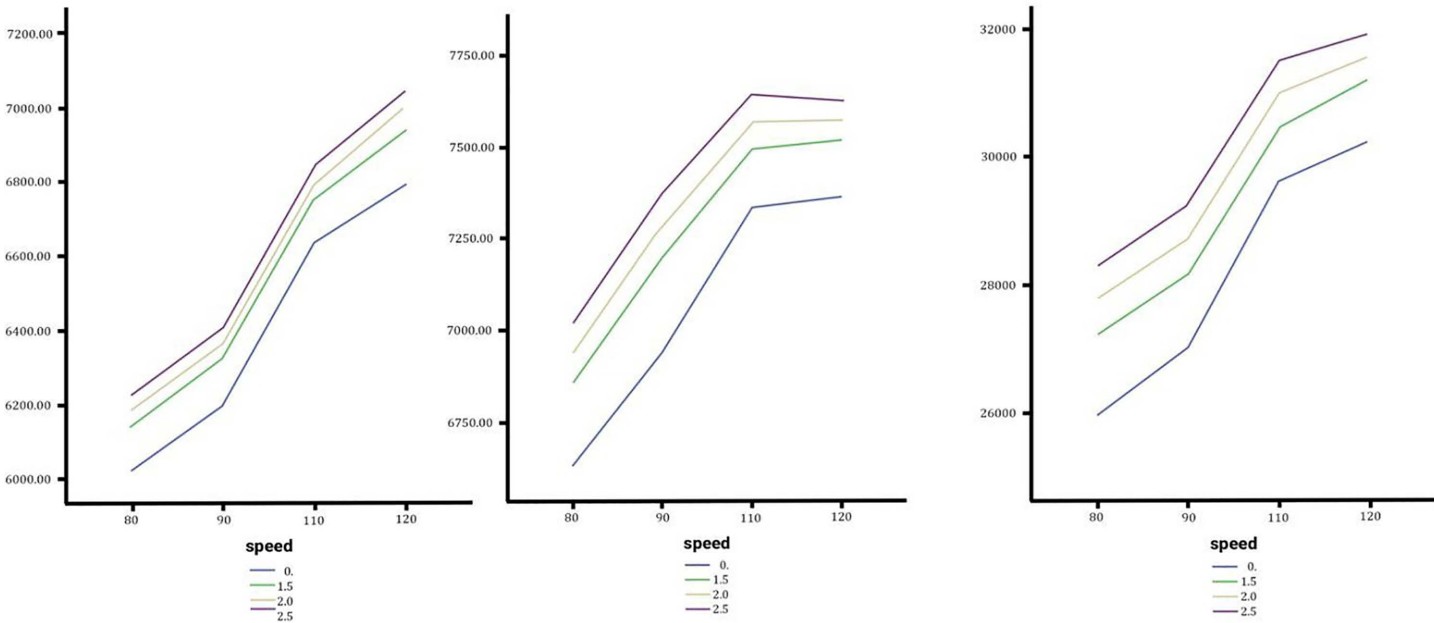

**Fig 12. In order from left to right the vertical force of the Sedan, SUV and Truck with no change, medium and high changes.**

**Table 3. changes in dynamic outputs for vehicles with changing the friction coefficient.**

|  | Friction change | Sedan | SUV | TRUCK |
|---|---|---|---|---|
| steering angle | From 0.9 to 0.5 | 0.51 | −5.68 | 1.49 |
|  | From 0.5 to 0.28 | 268.91 | 499.87 | 711.04 |
| slip angle | From 0.9 to 0.5 | 12.54 | 21.10 | 0.74 |
|  | From 0.5 to 0.28 | 928.98 | 1406.41 | 2293.07 |
| rollover angle | From 0.9 to 0.5 | −1.56 | −0.29 | −0.92 |
|  | From 0.5 to 0.28 | −12.76 | −19.85 | 50.22 |
| torsional moment angle | From 0.9 to 0.5 | 0.69 | 1.54 | 2.62 |
|  | From 0.5 to 0.28 | 27.93 | 448.02 | 1578.61 |
| torsional moment rate | From 0.9 to 0.5 | 0.19 | −1.36 | 5.7 |
|  | From 0.5 to 0.28 | 110.45 | 615.73 | 197.51 |
| Lateral acceleration | From 0.9 to 0.5 | −5.6 | −4.38 | −2.49 |
|  | From 0.5 to 0.28 | −25.24 | −34.67 | −16.72 |
| Vertical acceleration | From 0.9 to 0.5 | 0 | 0 | 11.18 |
|  | From 0.5 to 0.28 | 0 | 0 | 336.9 |
| horizontal force | From 0.9 to 0.5 | −10.61 | −16 | 4.20 |
|  | From 0.5 to 0.28 | −34.13 | −43.22 | −24.09 |
| Vertical force | From 0.9 to 0.5 | −1.35 | −2.67 | −0.52 |
|  | From 0.5 to 0.28 | −2.24 | −8.45 | −8.07 |

chassis-twist demand under low-grip lane-change conditions. Lateral acceleration decreases (more negative) for all vehicles as friction deteriorates, again most severely in SUVs and trucks, while vertical acceleration and forces climb for trucks, showing greater vertical load transfer sensitivity. Overall, the data highlights that friction loss introduces compounding instability, disproportionately affecting heavier and taller vehicles, and underscores the critical safety margin reduction when friction falls toward winter-like values ($\mu \approx 0.28$).

## Statistical significance analysis

To quantify the significance of parameter effects, a four-way ANOVA was conducted on the primary stability outputs across all 144 simulation scenarios (4 slopes × 4 speeds × 3 weather conditions × 3 vehicle types). Table 4 summarizes the key p-values, confirming that:

Speed and vehicle type exert the strongest main effects across all stability parameters (all $p < 0.01$).

Slope significantly influences rollover angle, yaw moment, and lateral acceleration ($p < 0.01$), with critical slope × speed interactions ($p < 0.01$) explaining trucks' instability at speeds above 90 km/h on 2.5% slopes.

Weather (friction) strongly affects slip angle ($p < 0.01$), amplifying slope sensitivity especially under snowy conditions.

High $R^2$ values (0.76–0.91) indicate excellent model fit, validating the observed trends as statistically robust.

These results quantitatively confirm the practical significance thresholds discussed in the study.

Table 5 shows that increasing the road's lateral slope from 0% to 2.5% produces generally moderate but consistent rises in dynamic response across all vehicles, with the largest sensitivities appearing in SUVs and especially trucks. Steering and slip angles increase only slightly for sedans, while SUVs and trucks show roughly double to triple the increments, indicating greater dependence on transverse slope for heavier and taller vehicles. Rollover angle exhibits the most pronounced growth: trucks experience a jump of nearly 160 units from 0% to 1.5%, far exceeding sedans and SUVs, demonstrating their higher rollover susceptibility even under small slope changes. Torsional moment angle and rate rise

**Table 4. ANOVA Results for Key Stability Parameters (p-values).**

| Factor | Rollover Angle | Slip Angle | Yaw Moment | Steering Angle | Lateral Acceleration |
|---|---|---|---|---|---|
| Slope (0–2.5%) | 0.002 *** | 0.124 | 0.001 *** | 0.032 ** | 0.005 *** |
| Speed (80–120 km/h) | 0.001 *** | 0.003 *** | 0.001 *** | 0.001 *** | 0.001 *** |
| Weather (Dry/Rain/Snow) | 0.015 ** | 0.001 *** | 0.028 ** | 0.156 | 0.002 *** |
| Vehicle Type | 0.001 *** | 0.001 *** | 0.001 *** | 0.001 *** | 0.001 *** |
| Slope × Speed | 0.001 *** | 0.087 * | 0.001 *** | 0.014 ** | 0.003 *** |
| Slope × Weather | 0.019 ** | 0.042 ** | 0.062 * | 0.234 | 0.023 ** |
| Speed × Weather | 0.004 *** | 0.001 *** | 0.017 ** | 0.098 | 0.001 *** |
| Model R² | 0.87 | 0.82 | 0.89 | 0.76 | 0.91 |

Notes:*** $p < 0.01$ (highly significant), ** $p < 0.05$ (significant), * $p < 0.10$ (marginal significance). Results derived from a full factorial design ANOVA (Slope × Speed × Weather × Vehicle Type).

**Table 5. change in dynamic outputs for different vehicles with increasing road lateral slope.**

| | lateral slope change | Sedan | SUV | TRUCK |
|---|---|---|---|---|
| steering angle | From 0.0 to 1.5 | 0.93 | 0.96 | 1.35 |
| | From 1.5 to 2.0 | 0.34 | 0.38 | 0.46 |
| | From 2.0 to 2.5 | 0.35 | 0.44 | 0.75 |
| slip angle | From 0.0 to 1.5 | 2.65 | 6.09 | 4.11 |
| | From 1.5 to 2.0 | 0.94 | 2.21 | 1.32 |
| | From 2.0 to 2.5 | 0.96 | 2.16 | 1.32 |
| rollover angle | From 0.0 to 1.5 | 41.44 | 58.72 | 159.88 |
| | From 1.5 to 2.0 | 11.06 | 14.07 | 20.50 |
| | From 2.0 to 2.5 | 9.95 | 12.32 | 17.32 |
| torsional moment angle | From 0.0 to 1.5 | 0.11 | 0.16 | 0.72 |
| | From 1.5 to 2.0 | 0.03 | 0.05 | 0.22 |
| | From 2.0 to 2.5 | 0.03 | 0.06 | 0.20 |
| torsional moment rate | From 0.0 to 1.5 | 0.55 | 1.21 | 1.83 |
| | From 1.5 to 2.0 | 0.14 | 0.53 | 0.41 |
| | From 2.0 to 2.5 | 0.13 | 0.97 | 0.39 |
| Lateral acceleration | From 0.0 to 1.5 | 1.21 | 3.7 | 1.13 |
| | From 1.5 to 2.0 | 0.40 | 1.24 | 0.93 |
| | From 2.0 to 2.5 | 0.38 | 1.41 | 1.00 |
| Vertical acceleration | From 0.0 to 1.5 | −0.04 | −0.08 | 93.62 |
| | From 1.5 to 2.0 | 0.03 | −0.06 | 73.97 |
| | From 2.0 to 2.5 | −0.04 | −0.08 | 38.36 |
| horizontal force | From 0.0 to 1.5 | 4.60 | 8.55 | 4.74 |
| | From 1.5 to 2.0 | 1.50 | 2.71 | 1.54 |
| | From 2.0 to 2.5 | 1.47 | 2.67 | 1.53 |
| Vertical force | From 0.0 to 1.5 | 1.99 | 3.41 | 4.77 |
| | From 1.5 to 2.0 | 0.69 | 1.13 | 1.99 |
| | From 2.0 to 2.5 | 0.69 | 1.14 | 1.98 |

gradually, again with trucks showing the greatest absolute changes, reflecting increased chassis twist demands. Lateral acceleration increases notably for SUVs and modestly for sedans and trucks, while vertical acceleration shows extremely large jumps for trucks—up to 93 units—signaling significant vertical load transfer on steeper crossfalls. Horizontal and vertical forces also rise with slope, with SUVs and trucks showing higher load variations than sedans. Overall, the data indicates that even small increases in road lateral slope amplify dynamic loads and stability-related responses, disproportionately affecting SUVs and especially trucks, which face the highest combined rollover, torsional, and load-transfer risks.

## Conclusion

This study focused on understanding the influence of bilateral transverse slopes on vehicle stability during lane-change maneuvers, addressing gaps in existing research by integrating diverse variables such as slope gradients, vehicle types, speeds, and weather conditions. Using advanced simulation tools like CarSim and TruckSim, findings revealed significant safety implications associated with slope-induced dynamics, particularly for heavier vehicles such as trucks. The research demonstrated that increasing slope gradients—up to the investigated maximum of 2.5%—and high speeds significantly amplify rollover risks, steering challenges, and instability, with adverse weather intensifying these effects. For instance, trucks exhibited the highest sensitivity, culminating in rollover and yaw issues under snowy conditions. While sedans and SUVs were comparatively stable, safety concerns remained notable beyond specific thresholds, reinforcing the importance of tailored road designs and advanced vehicular safety systems.

Key safety-related vehicle performance measures such as yaw stability, rollover risk, and lateral load transfer are critical for assessing crash risk because they directly influence a vehicle's ability to maintain control during dynamic maneuvers. Yaw stability, often represented by yaw rate and yaw moment, reflects how well a vehicle resists unwanted rotation around its vertical axis, which is essential to preventing spinouts or loss of directional control that can lead to severe crashes. Rollover risk, indicated by roll angle and lateral acceleration, reveals the propensity of a vehicle—especially those with higher centers of gravity like SUVs and trucks—to tip over during sharp lane changes or lateral disturbances, a major cause of serious injuries and fatalities. Lateral load transfer and slip angle characterize how forces shift across the tires during maneuvers and the likelihood of losing tire-road grip, which can result in sideslip or skidding. These dynamic measures serve as direct precursors to common crash types such as run-off-road, rollover, and multi-vehicle collisions, particularly under conditions of adverse weather, reduced friction, or elevated speeds, making them essential for predicting and mitigating crash likelihood. By monitoring and simulating these indicators, researchers and engineers can better understand vehicle stability thresholds, improve design standards, and tailor active safety systems to reduce crash risk effectively.

Analytical comparisons with prior studies reveal the comprehensive approach of this research. For example, [2,4,58] emphasized rollover risks in heavy vehicles under icy conditions but did not fully account for compound effects of slope percentages combined with weather and speed variations. Similarly, [20] explored road surface textures for improved traction in wet environments, offering practical measures against slippage but leaving critical interactions between slope geometry and vehicle dynamics underrepresented. Unlike these previous efforts, this study uniquely bridges gaps in understanding multidimensional factors, such as simultaneous analysis across three vehicle categories (sedans, SUVs, and trucks) and diverse weather scenarios.

By incorporating advanced simulations with real-world slope standards aligned with AASHTO guidelines (2018), this research provides insights that surpass the constrained analytical scope of earlier works, highlighting the compounded risks posed by bilateral transverse slopes under mixed driving conditions.

Moreover, this research contributes actionable recommendations to improve road and vehicle safety. For suburban roads with mixed traffic, minimizing slope gradients below 2% wherever possible could significantly reduce rollover risks. Additionally, enhancing Electronic Stability Control (ESC) systems to cater specifically to heavier vehicles, particularly trucks, offers long-term improvements in stability during lane-change maneuvers. For trucks on bilateral

transverse slopes, ESC systems should adopt slope-compensated intervention thresholds: ≤ 1.5% slopes maintain standard settings (lateral acceleration ≥ 0.4g, yaw rate ≥8°/s); 2.0–2.5% slopes require earlier activation at ≥0.35g lateral acceleration and ≥6°/s yaw rate to counter the observed 15–20% rollover angle increase at speeds above 90 km/h; snowy conditions (μ = 0.28) demand proactive intervention at ≥0.30g (truck rollover threshold) and ≥5°/s yaw rate. These simulation-derived parameters enable OEMs to recalibrate ESC algorithms for enhanced rollover prevention on drainage-optimized suburban roads without compromising normal driving performance. Practical measures such as adopting microtextured road designs or increasing friction coefficients in wet and snowy environments can further miti-gate risks. While the simulations utilized in this study provide robust findings, the research acknowledges its simplified assumptions, such as the absence of curved road sections or mixed traffic, which remain areas for future investigation. Compared to recent articles, this study offers a distinctive and comprehensive integration of road geometry, vehicle type, and weather diversity in the context of lane-changing dynamics on bilateral transverse slopes, areas often mar-ginalized in the latest simulation research. For instance, current studies such as [23] focus heavily on the behavioral and safety implications of lane changes under advanced driver assistance systems, emphasizing system response and the consistency of assisted maneuvers, but they rarely factor in complex road geometries or the compounding effect of adverse weather for multiple vehicle types [24].

utilize sophisticated simulation tools like PreScan and Simulink to analyze work zone layouts under autonomous driv-ing, identifying factors influencing traffic conflicts during lane changes but operating primarily on simplified, flat roadway models.

Unlike these, your paper's dynamic vehicle modeling encompasses a richer variety of environmental and operational parameters, offering nuanced safety insights specifically about the exacerbation of instability risks for trucks and under adverse weather. Additionally, studies using naturalistic data and artificial intelligence, such as by [14,54], provide robust analyses of the reciprocal relationships between congestion and lane-changing on real-world trajectories, yet stop short of multidimensional simulation under varying physical road conditions. Thus, your research stands out by meticulously bridg-ing simulation rigor with multidimensional real-world relevance, contributing actionable knowledge and practical design recommendations for the safe accommodation of diverse traffic on sloped roads in varied weather scenarios.

This study provides an exploratory yet comprehensive simulation-based assessment of vehicle stability on bilateral transverse slopes commonly used in suburban road drainage design. While real-world crash validation remains ongoing, the results identify key thresholds where stability degrades significantly, notably at slope levels of 2.5% or greater for heavy vehicles traveling above 90 km/h in low-friction conditions. Based on these findings, roadway engineers are advised to carefully consider limiting bilateral transverse slopes to below 2.5% in mixed traffic suburban settings or to supplement road design with advanced vehicle stability controls such as Electronic Stability Control (ESC). Future work will aim to integrate empirical crash and severity data to refine these guidance limits further and support targeted policy development for safer infrastructure planning.

For roadway engineers and policymakers, the key practical question is how much change in bilateral transverse slope significantly influences safety outcomes such as rollover risk and vehicle stability. Research indicates rollover thresholds for typical passenger vehicles occur around lateral accelerations of 0.8–1.2 g, with SUVs and trucks having lower thresh-olds due to higher centers of gravity. Slope-induced increases in rollover angles or lateral acceleration that approach or exceed these thresholds signify a meaningful safety concern. Studies show that for slopes exceeding approximately 2% to 2.5%, especially on wet or snowy pavements, rollover safety margins decrease markedly, increasing crash likelihood. Conversely, slope variations below roughly 1.5% tend to produce minimal changes in dynamic stability metrics and are generally considered safe in design standards. Therefore, slopes near or above 2.5% should be flagged as critical values where stability and rollover propensity rise significantly, warranting careful engineering consideration. This aligns with your simulation findings where rollover angles and yaw instability notably increase at high speeds on 2.5% slopes, especially for heavy vehicles, underscoring the need for balanced drainage design that does not compromise safety.

## Future research and limitation

The study highlights critical insights into the effects of bilateral transverse slopes on different type of vehicle safety during lane changes but identifies areas for further research. Future work should incorporate complex road features, such as curved alignments with superelevation and road shoulder interactions, to better reflect real-world conditions. Validating simulation findings with real-world traffic data, including mixed and diverse traffic dynamics and environmental factors, will enhance practical applicability. Additionally, investigating the impact of emerging vehicle technologies, like autonomous and electric, is essential as they may exhibit unique responses due to advanced suspension systems a driven controls. These efforts can provide comprehensive guidelines for safer road designs and adapt to evolvin vehicle behaviors.

This study focuses on detailed dynamic simulation of vehicle behavior on bilateral transverse slopes, leveraging the well-validated CarSim and TruckSim platforms to explore mechanistic relationships among slope, vehicle type, speed, and weather. Future research will extend this work by integrating empirical crash and severity data to triangulate safety-critical parameter thresholds and facilitate calibration of road design criteria.

Future extensions will incorporate variable driver inputs including emergency braking patterns, steering deviations, and adaptive speed control to capture realistic mixed-traffic interactions.

While this study models weather effects through realistic tire-road friction coefficients, future research will integrate visibility models and driver behavior adaptations to provide a more comprehensive assessment of adverse weather impacts on bilateral transverse slope stability.

## Supporting information

**S1 File. This file contains the complete minimal dataset and model input parameters used in this study, including vehicle geometric and mass properties (sedan, SUV, and truck), road geometry and lateral slope definitions, friction coefficients representing dry, rainy, and snowy conditions, simulation settings and the extracted output values used to generate all tables and figures in the manuscript.**
(ZIP)

## Acknowledgments

The authors acknowledge Imam Khomeini International University.

## Author contributions

**Conceptualization:** Mahdi Moharami, Ali Abdi Kordani.

**Data curation:** Mahdi Moharami, Akram Kohansal.

**Formal analysis:** Mahdi Moharami, Akram Kohansal.

**Funding acquisition:** Mahdi Moharami.

**Investigation:** Ali Abdi Kordani, Farzad Moradi.

**Methodology:** Ali Abdi Kordani, Akram Kohansal, Farzad Moradi.

**Project administration:** Ali Abdi Kordani.

**Resources:** Ali Abdi Kordani.

**Software:** Mahdi Moharami.

**Supervision:** Ali Abdi Kordani.

**Validation:** Ali Abdi Kordani, Akram Kohansal, Farzad Moradi.

**Visualization:** Farzad Moradi.

**Writing – original draft:** Farzad Moradi.

**Writing – review & editing:** Farzad Moradi.

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
