## [Decision Letter · Decision Letter 0]

17 Nov 2025

PONE-D-25-58368Integrated Multifactor Assessment of Road Geometry, Vehicle Types and Weather Diversity on Bilateral Transverse Slopes: Bridging Gaps in Dynamic ModelingPLOS ONE

Dear Dr. Abdi Kordani,

Thank you for submitting your manuscript to PLOS ONE. After careful consideration, we feel that it has merit but does not fully meet PLOS ONE’s publication criteria as it currently stands. Therefore, we invite you to submit a revised version of the manuscript that addresses the points raised during the review process.

We look forward to receiving your revised manuscript.

Kind regards,

Jinhao Liang

Academic Editor

PLOS ONE

Journal Requirements:

3. We note that your Data Availability Statement is currently as follows: All relevant data are within the manuscript and in Supporting Information files.

4. Please amend the manuscript submission data (via Edit Submission) to include author Akram kohansa.

5. We note that Figures 5 and 6 in your submission contain copyrighted images. All PLOS content is published under the Creative Commons Attribution License (CC BY 4.0), which means that the manuscript, images, and Supporting Information files will be freely available online, and any third party is permitted to access, download, copy, distribute, and use these materials in any way, even commercially, with proper attribution. For more information, see our copyright guidelines: http://journals.plos.org/plosone/s/licenses-and-copyright.

1. You may seek permission from the original copyright holder of Figures 5 and 6 to publish the content specifically under the CC BY 4.0 license.

Reviewers' comments:

Reviewer's Responses to Questions

**Comments to the Author**

1. Is the manuscript technically sound, and do the data support the conclusions?

Reviewer #1: Yes

Reviewer #2: Yes

2. Has the statistical analysis been performed appropriately and rigorously? 

Reviewer #1: Yes

Reviewer #2: Yes

3. Have the authors made all data underlying the findings in their manuscript fully available?

Reviewer #1: No

Reviewer #2: No

4. Is the manuscript presented in an intelligible fashion and written in standard English?

Reviewer #1: Yes

Reviewer #2: Yes

5. Review Comments to the Author

Reviewer #1: Thank you for invitation to review this article dear editor, below are my comments :

1. The motivation needs to be strengthened. The introduction asserts that roadway transverse slope affects safety, but the manuscript does not clearly explain why this specific geometry problem is urgent or understudied. The authors should contextualize why bilateral slope instability is a priority compared to other geometric factors (e.g., superelevation transitions, crossfall variation, or surface friction). Also,keywords were listed twice in the abstract such as “Keywords: accident severity, spatiotemporal analysis, machine learning, metaheuristic algorithm, random forest” and “Keywords: Bilateral transverse slope, lane change, vehicle dynamics model, road safety, simulation”, pick one of them in the revised manuscript.

2. Several recent studies that investigate the link between geometry, vehicle dynamic stability, and crash/injury outcomes are missing. The authors should incorporate work in accident severity modeling and spatiotemporal risk analysis to properly anchor the real-world safety relevance of the simulations such as https://doi.org/10.1080/23249935.2025.2516817, and https://doi.org/10.1016/j.aap.2025.108277 .

3. The manuscript states that vehicle behavior 'changes with slope,' but does not specify which outcomes matter most for safety and why. The authors should explicitly identify which performance measures (e.g., yaw stability, rollover risk, lateral load transfer) drive crash likelihood

4. Values selected in CarSim/TruckSim (vehicle mass, center of gravity, lane-change steering profile, speed input) are presented without referencing empirical data or prior experimental standards. The study should justify these values or provide sensitivity analysis to demonstrate robustness.

5. The paper presents only simulated results, and no comparison is made to observed crash patterns or empirical instability thresholds. Without any validation or triangulation, it is impossible to assess whether the findings have practical road design implications.

6. The results section lists trends (e.g., higher slope → increased rollover angle), but does not translate them into meaningful conclusions for roadway engineers and policy makers. The authors should explain how much slope change is significant enough to alter safety outcomes.

7. The manuscript claims that the results 'provide guidance for road design,' but no actionable thresholds, design limits, or policy recommendations are actually provided. The conclusion should be revised to reflect the exploratory nature of the study or include explicit guidance.

Reviewer #2: This manuscript focuses on the impact of bilateral transverse slopes on vehicle stability during lane changes, using multi-tool simulations to analyze multi-variable scenarios and address gaps in dynamic modeling. However, certain details need optimization to enhance rigor and practicality.

1. The simulated scenarios do not include curved roads or mixed traffic flows, which deviates from real-world road environments. It is recommended to supplement simulations of curved transverse slope scenarios to enhance the practicality of the research.

2. The assumptions about driver behavior are overly simplified, failing to account for real-world operations such as emergency braking and steering deviations. Variables for complex driving behaviors should be added to improve model authenticity.

3. The results section lacks significance analysis of parameter changes. It is suggested to supplement Analysis of Variance (ANOVA) or correlation analysis to clarify the statistical significance of the impacts of slope, speed, weather, and other factors on stability parameters.

4. Quantitative criteria for rollover thresholds of different vehicle types (especially trucks) are not clearly defined.

5. The analysis of weather factors only focuses on friction coefficients, ignoring key variables such as visibility.

6. Recommendations for ESC optimization for trucks are overly vague. Specific parameter adjustment plans (e.g., ESC intervention thresholds under different slopes) should be provided based on simulation data to enhance engineering operability.

6. PLOS authors have the option to publish the peer review history of their article (what does this mean? ). If published, this will include your full peer review and any attached files.

**Do you want your identity to be public for this peer review?** For information about this choice, including consent withdrawal, please see our Privacy Policy .

Reviewer #1: No

Reviewer #2: No

---

## [Author Response · Author response to Decision Letter 1]

27 Dec 2025

Dear reviewer

I am very grateful to the esteemed reviewer for reviewing the article and I am pleased to inform you that all the comments that were very helpful and improved the article have been fully edited, which have been fully marked up both in the article and in the file of 'Response to Reviewers" that is attached to your presence. Thank you again for your guidance that has strengthened the article.

---

## [Decision Letter · Decision Letter 1]

29 Dec 2025

Integrated Multifactor Assessment of Road Geometry, Vehicle Types and Weather Diversity on Bilateral Transverse Slopes: Bridging Gaps in Dynamic Modeling

PONE-D-25-58368R1

Dear Dr. Ali Abdi Kordani,

We’re pleased to inform you that your manuscript has been judged scientifically suitable for publication and will be formally accepted for publication once it meets all outstanding technical requirements.

Kind regards,

Jinhao Liang

Academic Editor

PLOS One

Additional Editor Comments (optional):

Reviewers' comments:

Reviewer's Responses to Questions

**Comments to the Author**

1. If the authors have adequately addressed your comments raised in a previous round of review and you feel that this manuscript is now acceptable for publication, you may indicate that here to bypass the “Comments to the Author” section, enter your conflict of interest statement in the “Confidential to Editor” section, and submit your "Accept" recommendation.

Reviewer #1: All comments have been addressed

Reviewer #2: All comments have been addressed

2. Is the manuscript technically sound, and do the data support the conclusions?

Reviewer #1: (No Response)

Reviewer #2: Yes

3. Has the statistical analysis been performed appropriately and rigorously? 

Reviewer #1: (No Response)

Reviewer #2: Yes

4. Have the authors made all data underlying the findings in their manuscript fully available?

Reviewer #1: (No Response)

Reviewer #2: Yes

5. Is the manuscript presented in an intelligible fashion and written in standard English?

Reviewer #1: (No Response)

Reviewer #2: Yes

6. Review Comments to the Author

Reviewer #1: (No Response)

Reviewer #2: All my comments have bee addressed and the manuscript can be accepted now. Thanks for your revision.

7. PLOS authors have the option to publish the peer review history of their article (what does this mean? ). If published, this will include your full peer review and any attached files.

**Do you want your identity to be public for this peer review?** For information about this choice, including consent withdrawal, please see our Privacy Policy .

Reviewer #1: No

Reviewer #2: **Yes:** Gen Li

---

## [Editor Report · Acceptance letter]

PONE-D-25-58368R1

PLOS One

Dear Dr. Abdi Kordani,

I'm pleased to inform you that your manuscript has been deemed suitable for publication in PLOS One. Congratulations! Your manuscript is now being handed over to our production team.

Kind regards,

on behalf of

Dr. Jinhao Liang

Academic Editor

PLOS One